# Dysregulation of LINC00324 promotes poor prognosis in patients with glioma

Xin Jin[1☉], Jiandong Zhu[1☉], Haoyun Yu[2], Shengjun Shi[3], Kecheng Shen[1], Jingyu Gu[1], Ziqian Yin[1], Zhengquan Yu[1]*, Jiang Wu[1]*

1 Department of Neurosurgery, The First Affiliated Hospital of Soochow University, Suzhou, China, 2 Suzhou Medical College, Soochow University, Suzhou, China, 3 Department of Neurosurgery, The Shengze Hospital Affiliated with Nanjing Medical University, Suzhou, China

☉ These authors contributed equally to this work.
* szjiangwu@163.com (JW); zhengquan.yu@neurosci.com.cn (ZY)

## Abstract

### Background

LINC00324 is a long-stranded non-coding RNA, which is aberrantly expressed in various cancers and is associated with poor prognosis and clinical features. It involves multiple oncogenic molecular pathways affecting cell proliferation, migration, invasion, and apoptosis. However, the expression, function, and mechanism of LINC00324 in glioma have not been reported.

### Material and methods

We assessed the expression of LINC00324 of LINC00324 in glioma patients based on data from The Cancer Genome Atlas (TCGA) and Genotype-Tissue Expression (GTEx) to identify pathways involved in LINC00324-related glioma pathogenesis.

### Results

Based on our findings, we observed differential expression of LINC00324 between tumor and normal tissues in glioma patients. Our analysis of overall survival (OS) and disease-specific survival (DSS) indicated that glioma patients with high LINC00324 expression had a poorer prognosis compared to those with low LINC00324 expression. By integrating clinical data and genetic signatures from TCGA patients, we developed a nomogram to predict OS and DSS in glioma patients. Gene set enrichment analysis (GSEA) revealed that several pathways, including JAK/STAT3 signaling, epithelial-mesenchymal transition, STAT5 signaling, NF-κB activation, and apoptosis, were differentially enriched in glioma samples with high LINC00324 expression. Furthermore, we observed significant correlations between LINC00324 expression, immune infiltration levels, and expression of immune checkpoint-related genes (HAVCR2: r = 0.627, P = 1.54e-77; CD40: r = 0.604, P = 1.36e-70; ITGB2: r = 0.612, P = 6.33e-7; CX3CL1: r = -0.307, P = 9.24e-17). These findings highlight the potential significance of LINC00324 in glioma progression and suggest avenues for further research and potential therapeutic targets.

**Data Availability Statement:** The data underlying the results presented in the study are available from TCGA database, which is free for anyone to download and analyze at https://portal.gdc.cancer.gov/analysis_page?app=Downloads.

**Funding:** This work was supported by the Medical and Health Science and Technology Innovation Project of the Suzhou Health Commission [grant number: SKY2022002]. The funders had no role in study design, data collection and analysis, decision to publish, or preparation of the manuscript.

**Competing interests:** The authors have declared that no competing interests exist.

## Conclusion

Indeed, our results confirm that the LINC00324 signature holds promise as a prognostic predictor in glioma patients. This finding opens up new possibilities for understanding the disease and may offer valuable insights for the development of targeted therapies.

## Background

Gliomas are common malignant primary brain tumors, accounting for 50–60% of all brain tumors [1, 2]. Once diagnosed, maximal surgical resection followed by adjuvant radiotherapy or chemotherapy is the standard for patients with gliomas [3, 4]. Although traditional clinical influences are still used to predict prognosis and guide treatment [5], such as tumor size, location, age, and clinical and pathological grade, they are less effective, with a 5-year survival rate of less than 10% [6]. Advances in genomic studies have greatly improved our understanding of the underlying molecular mechanisms of glioma, highlighting the clinical application of molecular biomarkers in diagnosis and prognosis to enhance the survival of glioma patients.

Human genome sequencing technologies have shown that most of the human genome is transcribed with RNAs that have no protein-coding capacity, and they are called non-coding RNAs [7, 8], including microRNAs, circular RNAs, long-stranded non-coding RNAs (lncRNAs) and pseudogenes [9]. lncRNAs regulate various biological behaviors, including cell differentiation, proliferation, apoptosis, and metastasis, and their crucial role in the generation and pathogenesis of cancer has been demonstrated [10]. In the past few years, lncRNAs have received extensive attention in glioma research and are considered to be of great value as diagnostic biomarkers and therapeutic targets [11, 12]. Han et al. compared the lncRNA expression profiles of glioblastoma and normal brain tissues and found that lncRNAs targeting growth factor-related genes play an essential role in the regulation of glioma signaling pathways by gene network analysis [13]. It has also been found that lncRNAs such as CRNDE and HOTAIRM1 are closely associated with the malignancy of astrocytes [14]. In addition, several prognostic models for lncRNAs have been identified in gliomas.

LINC00324, also known as c17orf44, is located on human chromosome 17p13.1 and is a 2082-bp long intervening/intergenic non-coding RNA [15]. It is aberrantly expressed in various types of cancer and associated with poor prognosis and clinical features. LINC00324 involves multiple oncogenic molecular pathways affecting cell proliferation, migration, invasion, and apoptosis [16–18]. However, the expression, function, and mechanism of LINC00324 in gliomas have yet to be explored. In this study, we analyzed the expression and clinical relevance of LINC00324 in glioma. We investigated the role of LINC00324 in glioma cell proliferation and tumorigenesis. We further attempted to elucidate the molecular mechanisms associated with LINC00324-dependent phenotypes. We also constructed a reliable prognostic model to determine the prognosis of glioma patients.

## Material and methods

### Data collection and processing

We collected the mRNA expression profiles and clinical data of 699 glioma samples and 5 normal samples from the Cancer Genome Atlas (TCGA) database(https://tcga-data.nci.nih.gov). The other 1152 normal samples were obtained from the Genotype-Tissue Expression (GTEx) database (https://gtexportal.org/). The level 3 HTSeq-FPKM format data were normalized as

transcripts per million reads. The clinicopathological data included age, gender, World Health Organization(WHO) grade, histological type, Isocitrate Dehydrogenase 1(IDH1) status, 1p/19q codeletion status, and primary therapy outcome. This is a bioinformatics study. The Ethics Committee of First Affiliated Hospital of Soochow University has confirmed that no ethical approval is required.

## Survival analysis

Survival analysis was performed using the Kaplan-Meier method along with the log-rank test, with the median expression level of LINC00324 set as the cut-off value. Univariate and multivariate Cox regression analyses were employed to evaluate the impact of clinical variables on patient outcomes. In the multivariate analysis, variables with a significance level of $P < 0.1$ from the univariate analysis were included. Visualization of the results was accomplished using the "ggplot2" R package to generate a forest plot. Additionally, the developed signature enabled estimation of 1-year, 3-year, and 5-year survival rates using the nearest neighbor estimation method (NNE) [19]. The predictive power of the signature was assessed by constructing receiver operating characteristic (ROC) curves, utilizing the "timeROC" R package.

## Establishment and evaluation of the nomogram

In the present study, LINC00324 expression and the clinical features were used to build a nomogram, which is a practical and convenient approach for estimating the overall survival in individual patients [20]. The calibration curve and area under the ROC curve(AUC) were performed to verify and evaluate the prediction accuracy of the nomogram.

## Gene network and enrichment analysis

To explore the potential functions associated with LINC00324 expression, we employed Gene Set Enrichment Analysis (GSEA). The samples were categorized into high and low LINC00324 expression groups, serving as a training set. This allowed us to identify significant differences in survival and potential functions. The gene set h.all.v7.5 was selected from the msigdb collection as an annotated gene set [21]. We considered gene sets with a false discovery rate $< 0.25$ and an adjusted P-value $< 0.05$ as significantly enriched using the GMT (genes as reference gene set) method. Multiple genome alignments were conducted for each analysis. The enriched pathways for each phenotype were ranked based on normalized enrichment scores and adjusted P-values. Enrichment analysis was performed using Gene Ontology (GO) and Kyoto Encyclopedia of Genes and Genomes (KEGG) as references, facilitated by the R package "clusterProfiler" [22]. Multiple corrections were applied using the Benjamini-Hochberg method, with an adjusted P-value $< 0.05$ deemed significant.

## LINC00324 expression association with immune cells

Literature analysis identified 24 immune cell marker genes, including innate immune cells (dendritic cells[DCs], immature DCs [iDCs], activated DCs [aDCs], eosinophils, mast cells, macrophages, natural killer cells [NKs], NK CD56dim cells, NK CD56bright cells, and neutrophils) and adaptive immune cells (B, T helper 1 [Th1], Th2, T gamma delta [Tgd], CD8+ T, T central memory [Tcm], T effector memory [Tem], and T follicular helper [Tfh] cells) [23]. To assess the infiltration of immune cells in glioma, we conducted single-sample gene set enrichment analysis (ssGSEA). The degree of correlation between LINC00324 and immune cells was analyzed using the Spearman correlation method. Additionally, the Wilcoxon rank sum test was employed to compare the infiltration levels of these cells between samples with high and

low expression of LINC00324. Another aspect of our investigation focused on primary immune checkpoints (ICKs), as the expression levels of immune checkpoint-related genes have been linked to the therapeutic response to immune checkpoint inhibitors [24, 25]. We utilized Pearson correlation analysis to examine the association between ICK and LINC00324 expression.

## Protein–protein interaction network construction and analysis

To construct protein-protein interaction (PPI) networks, we utilized the Retrieval of Interacting Genes (STRING) database search tool (http://string-db.org/). The generated PPI networks were then visualized using Cytoscape, a bioinformatics analysis software (version 3.9.1), known for its capability to visualize and analyze complex biological networks.

## Statistical analysis

We performed a descriptive statistical analysis of glioma patients in TCGA. Wilcoxon rank sum test and logistic regression were used to analyze the association between clinical factors and LINC00324. We used the chi-square test for categorical variables and the t-test for statistical analysis of numerical variables. And all statistical analyses were performed using R version 4.1.1. P values were two-sided, and we considered a level of P value less than 0.05 to be statistically significant.

## Results

### The expression level of LINC00324 in gliomas and the association with clinicopathologic factors

Table 1 summarizes the TCGA dataset of 699 tumors. Our study cohort included 349 LINC00324 low-expressing samples and 350 LINC00324 high-expressing samples. All patients were diagnosed with glioblastoma (GBM) or low-grade glioma (LGG). Histograms showing differences in LINC00324 expression between normal and tumor samples (Fig 1A). The results indicated that the expression of LINC00324 was significantly different between normal and tumor samples and may play a crucial role in regulating cancer development. Univariate analysis using logistic regression showed that LINC00324 expression as a variable was associated with clinicopathological factors used to determine poor prognosis (Table 2).

The expression of LINC00324 in GBM and LGG exhibited significant correlations with various clinicopathological factors. Age showed a significant association (OR 2.174; 95% CI 1.489–3.203, >60 vs. ≤60), as did WHO grade (OR 3.118; 95% CI 2.225–4 and G249 vs. G3), status of Relative IDH1 (OR 0.179; 95% CI 0.125–0.253, mutant [Mut] vs. wild-type [WT]), 1p/19q co-deletion status (OR 0.071; 95% CI 0.041–0.116, codel), and primary treatment outcome (OR 1.708; 95% CI 1.126–2.621, progressive disease [PD], stable disease [SD], and partial response [PR] versus complete response [CR]). These findings indicate that patients with high LINC00324 mRNA expression tended to be at a more advanced stage and more likely to experience disease progression.

Furthermore, analysis of the TCGA database and GTEx database revealed significantly elevated expression of LINC00324 in GBM and LGG tissues compared to normal tissues (P<0.05). Receiver operating characteristic analysis demonstrated the potential of LINC00324 expression as a differentiating factor, with an area under the curve (AUC) of 0.873 for LINC00324-expressing gliomas compared to normal tissues (Fig 1B).

### Prognostic value of LINC00324 transcription expression in gliomas

Prognostic statistics were extracted from the TCGA database, as depicted in Fig 2A and 2D, revealing a significant association between high LINC00324 expression in gliomas and shorter overall survival (OS) (P<0.001) and disease-specific survival (DSS) (P<0.001). We further

**Table 1. Characteristics of patients with glioma in TCGA\*.**

| Characteristics | Low expression of LINC00324 | High expression of LINC00324 | *P* value |
|---|---|---|---|
| n | 349 | 350 | |
| Age, median(IQR) | 42 (33, 53) | 51 (36, 62) | < 0.001 |
| Gender, n (%) | | | 0.7350 |
| Female | 151 (21.6%) | 147 (21%) | |
| Male | 198 (28.3%) | 203 (29%) | |
| WHO grade, n (%) | | | < 0.001 |
| G2 | 146 (22.9%) | 78 (12.2%) | |
| G3 | 133 (20.9%) | 112 (17.6%) | |
| G4 | 27 (4.2%) | 141 (22.1%) | |
| Histological type, n (%) | | | < 0.001 |
| Astrocytoma | 84 (12%) | 112 (16%) | |
| Glioblastoma | 27 (3.9%) | 141 (20.2%) | |
| Oligoastrocytoma | 80 (11.4%) | 55 (7.9%) | |
| Oligodendroglioma | 158 (22.6%) | 42 (6%) | |
| IDH1 status, n (%) | | | < 0.001 |
| Wildtype | 61 (8.9%) | 185 (26.9%) | |
| Mutation | 283 (41.1%) | 160 (23.2%) | |
| 1p/19q codeletion, n (%) | | | < 0.001 |
| Non-codel | 196 (28.3%) | 324 (46.8%) | |
| Codel | 153 (22.1%) | 19 (2.7%) | |
| Primary therapy outcome, n (%) | | | < 0.01 |
| CR | 97 (20.9%) | 43 (9.2%) | |
| PD | 53 (11.4%) | 59 (12.7%) | |
| PR | 44 (9.5%) | 21 (4.5%) | |
| SD | 86 (18.5%) | 62 (13.3%) | |

\*IQR, interquartile range; G2, grade 2; G3, grade 3; G4, grade 4; CR, complete remission; PD, progressive disease; PR, partial remission; SD, stable disease.

utilized the expression level of LINC00324 to predict the 1-, 3-, and 5-year OS rates, as illustrated in Fig 2B. The corresponding area under the curve (AUC) values for the 1-year, 3-year, and 5-year OS ROC curves were reported as 0.740, 0.744, and 0.686, respectively. Similar analyses were conducted for the DSS outcome, with reported AUC values of 0.757, 0.755, and 0.709 for the 1-year, 3-year, and 5-year DSS ROC curves (Fig 2E). Additionally, time-dependent AUC curves were plotted to illustrate the change in OS and DSS over a follow-up period of 1 to 10 years (Fig 2C and 2F).

To enhance the development of predictive models, we integrated clinical and genetic data of TCGA patients and constructed nomograms using multivariate Cox regression models. These nomograms were designed to provide a comprehensive assessment of prognostic outcomes in terms of OS and DSS (Fig 3). Notably, calibration plots demonstrated good agreement between the estimated OS or DSS and observed OS or DSS rates at 1-, 3-, and 5-year survival intervals (Fig 3C and 3F).

## Differential expression of LINC00324 in different clinicopathological subgroups of gliomas

As shown in Fig 4, the expression levels of LINC00324 varied in different clinicopathological gliomas subgroups, except for the primary therapy outcome group, which did not show any

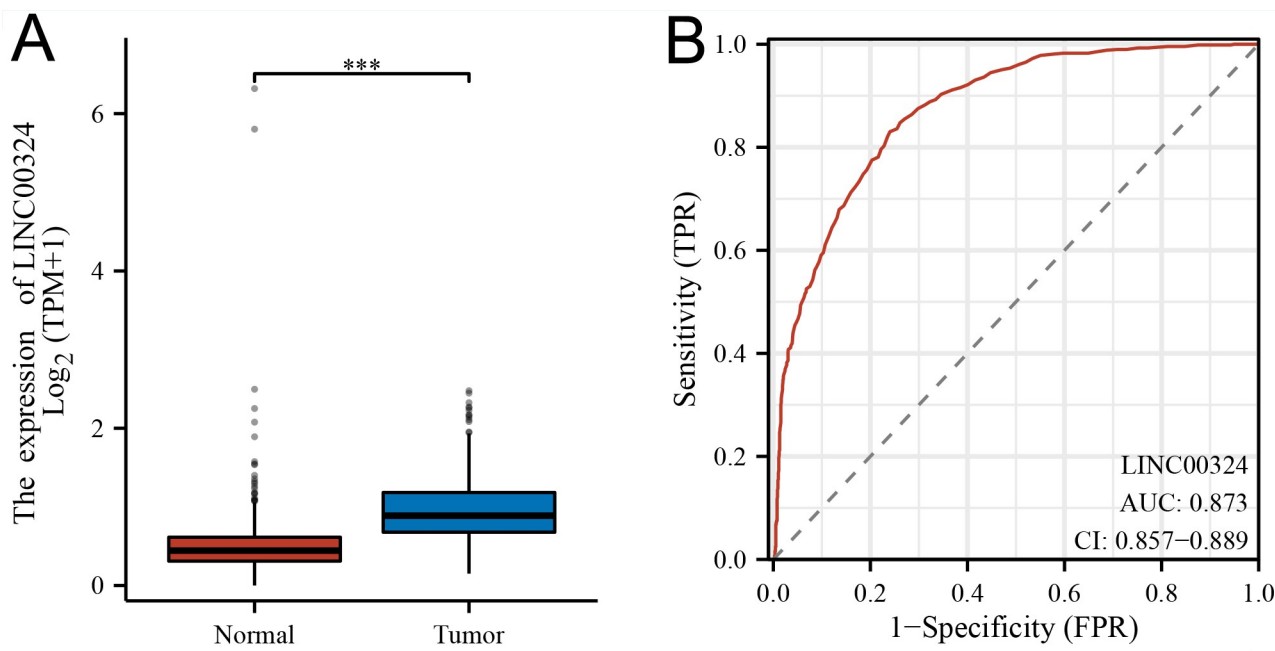

**Fig 1. Expression of LINC00324 in glioma.** (A) The expression of LINC00324 in glioma and normal tissues(*, P<0.05). (B) Order of receiver operating characteristic curves for LINC00324 expression in normal and glioma tissues. TPR, true positive rate; FPR, false positive rate.

statistical difference(Fig 4F). Specifically, with the increase of age, the expression of LINC00324 increased. There was an upward trend of LINC00324 expression regardless of tumor stage. The expression of LINC00324 in glioblastoma was significantly higher than that in other pathological types, as well as in IDH1-WT and 1p19q non-codel groups(Fig 4A–4E). These results suggested that the expression of LINC00324 may be correlated with disease grade and the molecular typing of gliomas. And the high expression of LINC00324 predicted a poor prognosis regarding the gliomas.

### Enrichment analysis of LINC00324 functional networks in gliomas

In order to investigate the role of LINC00324 in glioma development, we conducted RNAseq gene expression analysis to compare the gene expression profiles between the high and low expression groups of LINC00324. Combined plots were used to illustrate the expression of

**Table 2. Association between LINC00324 expression and clinicopathologic parameters by Logistic regression*.**

| Characteristics | Total(N) | Odds Ratio(OR) | *P* value |
|---|---|---|---|
| Age (>60 vs. ≤60) | 696 | 2.174 (1.489–3.203) | <0.001 |
| Gender (Male vs. Female) | 696 | 1.048 (0.776–1.416) | 0.759 |
| WHO grade (G3&G4 vs. G2) | 635 | 3.118 (2.225–4.399) | <0.001 |
| Histological type (Astrocytoma & Oligoastrocytoma & Oligodendroglioma vs. Glioblastoma) | 696 | 0.117 (0.073–0.181) | <0.001 |
| IDH1 status (Mutation vs. Wildtype) | 686 | 0.179 (0.125–0.253) | <0.001 |
| 1p/19q codeletion (codel vs. non-codel) | 689 | 0.071 (0.041–0.116) | <0.001 |
| Primary therapy outcome (PD&SD&PR vs. CR) | 462 | 1.708 (1.126–2.621) | 0.013 |

* G2, grade 2; G3, grade 3; G4, grade 4; CR, complete remission; PD, progressive disease; PR, partial remission; SD, stable disease.

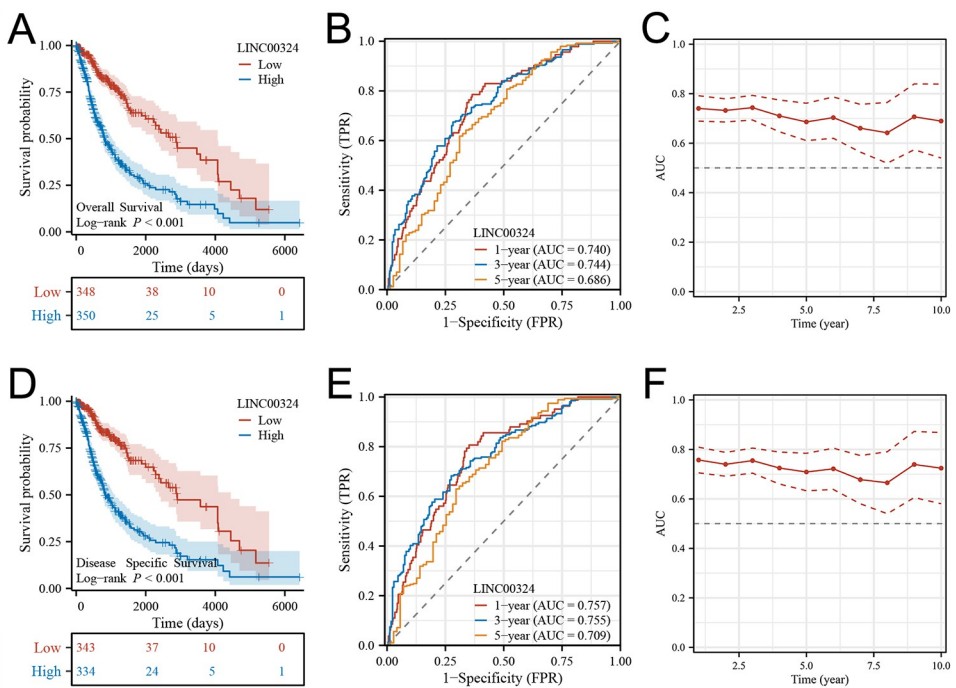

**Fig 2. Clinical relevance of LINC00324 in glioma patients.** For OS outcome, (A) Kaplan-Meier plot of the LINC00324 signature and overall survival, (B) ROCs for 1-, 3- and 5-year survival prediction, and (C) AUCs for predicting 1–10 year survival. For DSS outcome, (D) Kaplan-Meier plot of the LINC00324 signature and disease-specific survival. (E) ROCs for 1-, 3- and 5-year disease-specific survival prediction, (F) AUCs for predicting 1–10 years disease-specific survival. OS, overall survival; DSS, disease-specific survival; ROC, receiver operating characteristic curve; AUC, area under ROC curve; TPR, true positive rate; FPR, false positive rate.

LINC00324 in normal and tumor samples, revealing significant differences between the two (Fig 4). A total of 2538 differentially expressed genes (DEGs) were identified in the LINC00324 high expression group, including 1857 up-regulated genes (fold change >1.5) and 681 down-regulated genes (fold change <-1.5). Volcano plots were constructed to visualize the differences among DEGs (Fig 5).

To gain further insight into the biological pathways associated with glioma pathogenesis and stratified by LINC00324 expression level, we performed Gene Set Enrichment Analysis (GSEA). The GSEA enrichment map indicated that gene signatures related to the JAK/STAT3 signaling pathway, epithelial-mesenchymal transition, STAT5 signaling pathway, NF-κB activation, and apoptosis were enriched in patients with high expression of LINC00324 (Fig 6). These findings suggest that LINC00324 may play a role in glioma development by involving multiple pathways.

To elucidate the biological activities of LINC00324, we conducted Gene Ontology (GO) and Kyoto Encyclopedia of Genes and Genomes (KEGG) pathway analyses. GO analysis revealed that LINC00324 was predominantly associated with biological processes such as humoral immune response, lymphocyte-mediated immunity, and adaptive immune response. Moreover, it was found to be involved in cellular components like the extracellular plasma membrane and synaptic membrane, and exhibited molecular functions including antigen binding, passive transmembrane transporter activity, and channel activity (Fig 7A–7C). In the KEGG pathway analysis, LINC00324 was primarily associated with neuroactive ligand-receptor interactions, cytokine-cytokine receptor interactions, and systemic lupus erythematosus

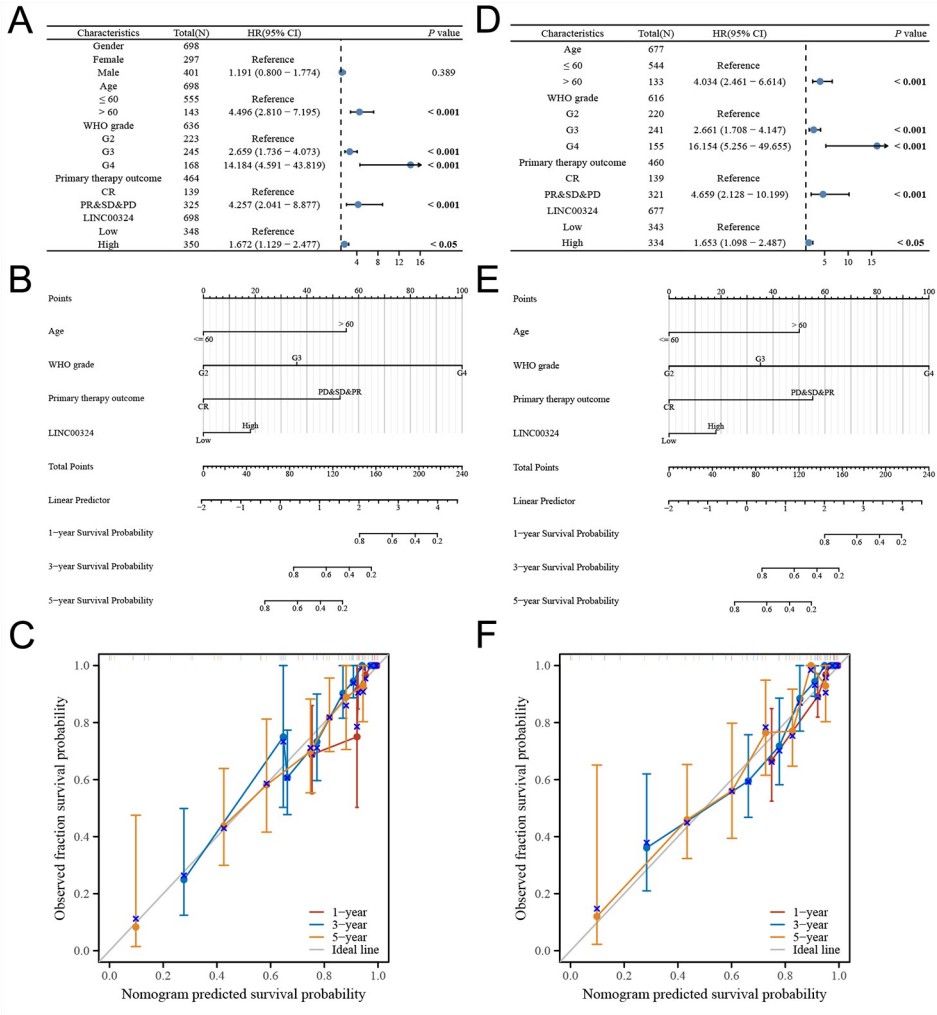

**Fig 3. Nomogram development and validation.** For OS(A) and DSS(D), hazard ratios and P-value of the constituents involved in multivariate Cox regression considering clinical information and prognostic LINC00324 in glioma. Nomogram to predict the 1-, 3- and 5-year OS(B) and DSS(E) rate of glioma patients. Calibration curve for the OS(C) and DSS(F) nomogram model in glioma. A dashed diagonal line represents the ideal nomogram. OS, overall survival; DSS, disease-specific survival; G2, grade 2; G3, grade 3; G4, grade 4; CR, complete remission; PD, progressive disease; PR, partial remission; SD, stable disease.

(Fig 7D). The network depicting the potential co-expression of LINC00324 with other genes within the DEGs associated with LINC00324 is presented in Fig 8.

## Correlation between expression of LINC00324 and immune infiltration levels in gliomas

To investigate whether LINC00324 may influence immune cell recruitment in the tumor microenvironment and subsequently impact glioma prognosis, we conducted an analysis of the relationships between LINC00324 expression and immune infiltration in gliomas. This analysis was performed using single-sample Gene Set Enrichment Analysis (ssGSEA).

The results demonstrated that LINC00324 expression exhibited a positive correlation with the abundance of innate immune cells, such as macrophages, neutrophils, eosinophils, and natural killer (NK) cells. Conversely, there was a negative correlation between LINC00324

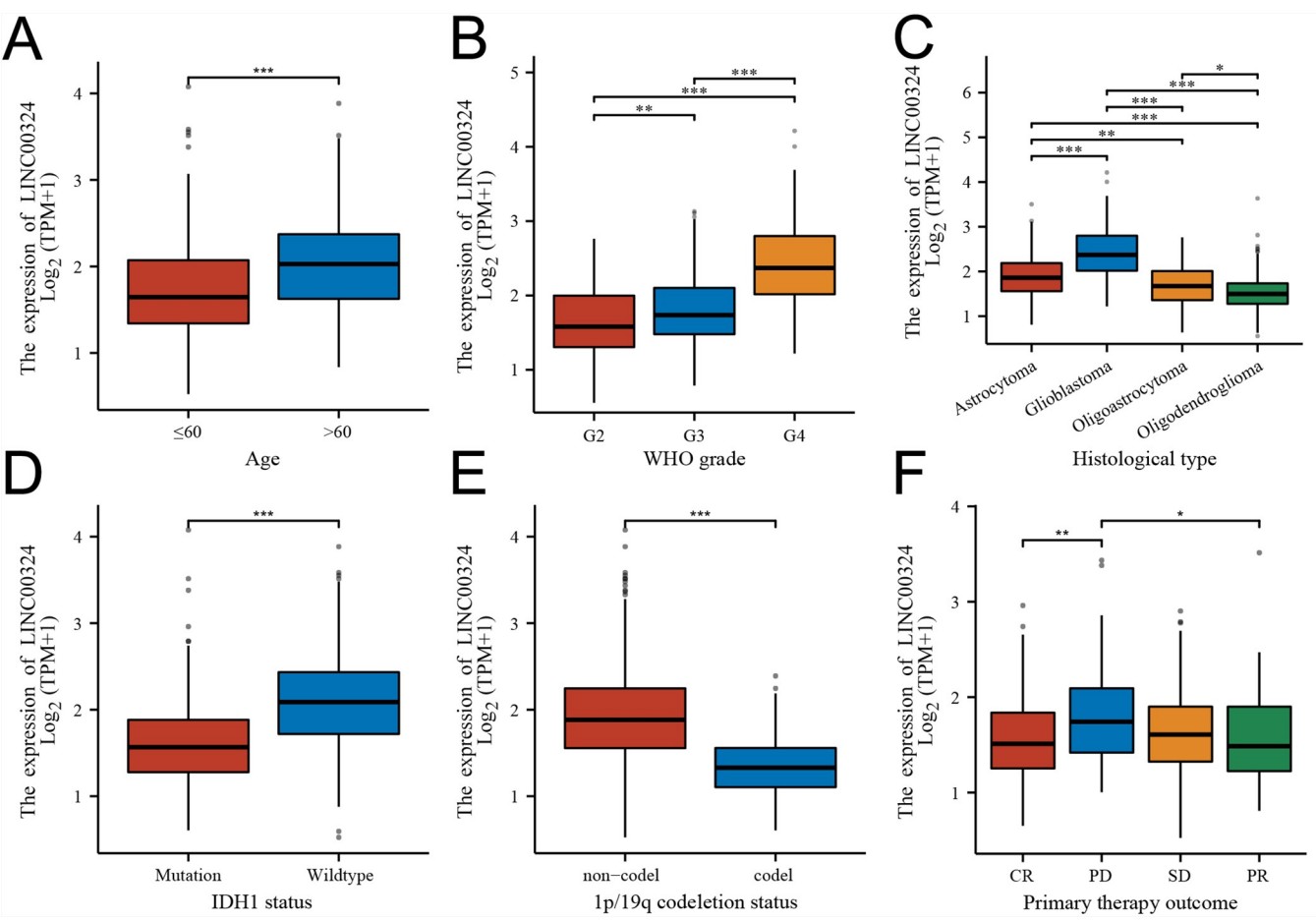

**Fig 4.** Expression of LINC00324 in a different age stage(A), pathologic stage(B), histological type(C), IDH1 status(D), 1p/19q codeletion status(E), and primary therapy outcome(F) of glioma patients, respectively. G2, grade 2; G3, grade 3; G4, grade 4; CR, complete remission; PD, progressive disease; PR, partial remission; SD, stable disease(*, P<0.05;**, P<0.01;***, P<0.001).

expression and the abundance of adaptive immunocytes, including Tcm, Tgd, Tem, and CD8 + T cells. However, even in cases with high LINC00324 expression, the recruitment of adaptive immune cells, such as B cells, Th1 cells, and Th2 cells, was still observed (Fig 9). These findings suggest that LINC00324 may have a complex impact on immune cell infiltration in the glioma microenvironment.

We also analyzed the correlation between the expression of LINC00324 and immune check-point-related genes by Pearson correlation (S1 Table) and found that the LINC00324 expression was negatively correlated with CX3CL1(r = -0.307, P = 9.24e-17), IFNA2(r = -0.174, P = 3.60e-06), and ADORA2A(r = -0.185, P = 8.86e-07), and positively correlated with HAVCR2(r = 0.627, P = 1.54e-77), CD40(r = 0.604, P = 1.36e-70), and ITGB2(r = 0.612, P = 6.33e-73) (Fig 10).

## Discussion

In the past decades, a growing body of evidence has revealed the critical role of lncRNAs in human diseases, especially cancer [26–28]. In particular, in gliomas, multiple lncRNAs are aberrantly expressed; their aberrant expression correlates with the malignancy of gliomas. These molecules are involved in glioma progression by regulating cell proliferation, apoptosis, differentiation, and stress response to hypoxia [29–31]. Therefore, a better understanding of

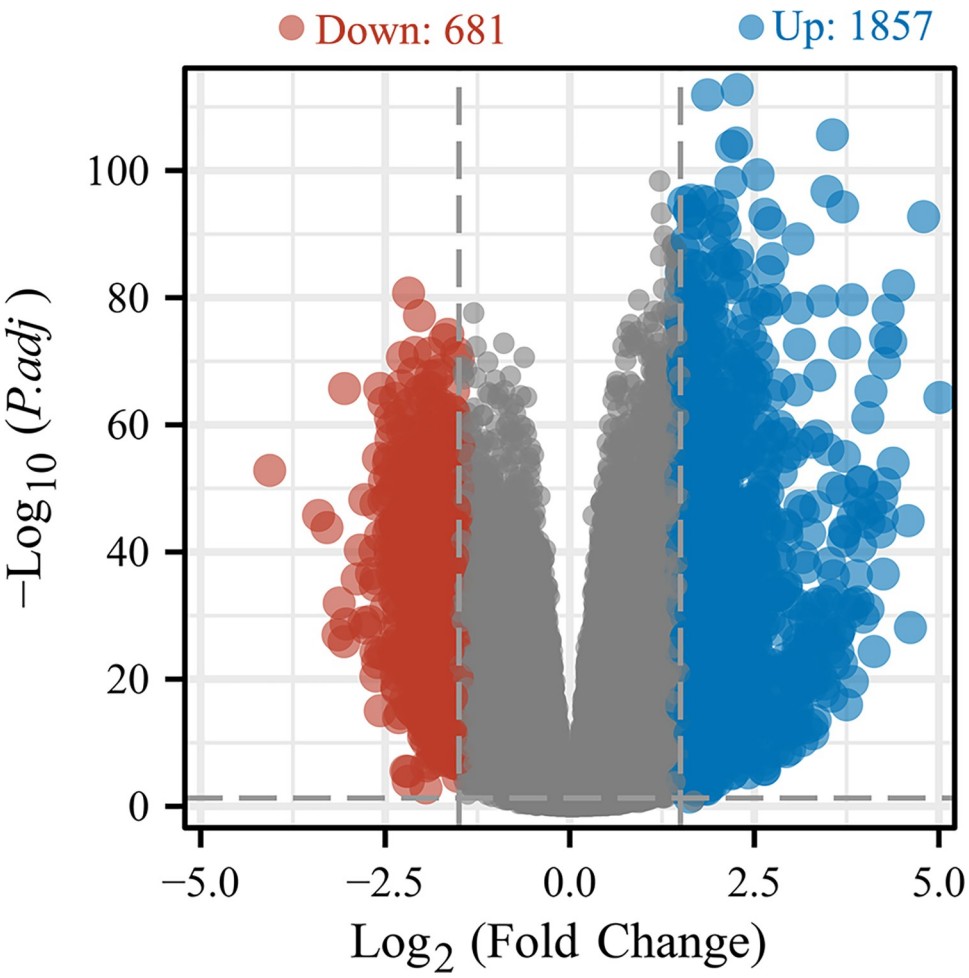

**Fig 5. Differential expression analysis of high- and low-expression LINC00324 groups in the volcano map.**

the lncRNA activity associated with gliomas is of great importance in finding potential targets for diagnosing and treating this malignant tumor. In this study, we performed a comprehensive and detailed analysis of LINC00324 expression in glioma patients to explore its association with clinicopathological features and survival and its role in cancer progression. Our results suggest that LINC00324 expression is abnormally prominent in tumor samples and regulates cancer progression through biological responses such as apoptosis. Cox regression analysis showed that clinical stage and primary treatment outcome were independent prognostic factors. Here, we observed that LINC00324 was upregulated in grade 4 compared to grade 3, grade 3 compared to grade 2, and grade 4 compared to grade 1 gliomas. Moreover, LINC00324 expression was generally upregulated in IDH1 wild-type glioma patients with worse prognosis, suggesting that LINC00324 may be involved in the mechanism of glioma IDH mutation and tumor metabolic reprogramming [32, 33]. Although some lncRNAs associated with IDH1 mutations (HOTAIRM1, NCRNA00173, MIR155HG, etc.) are involved in various tumor-associated cellular processes [34]. However, the exact mechanism of LINC00324 upregulation in malignant tissues has not been determined. A study in patients with acute myeloid leukemia suggested that hypomethylation of LINC00324 may be the mechanism of its upregulation [35]. It has been shown that miR-375 expression is downregulated in gliomas and that high expression of miR-375 can target and inhibit RWDD3 expression and

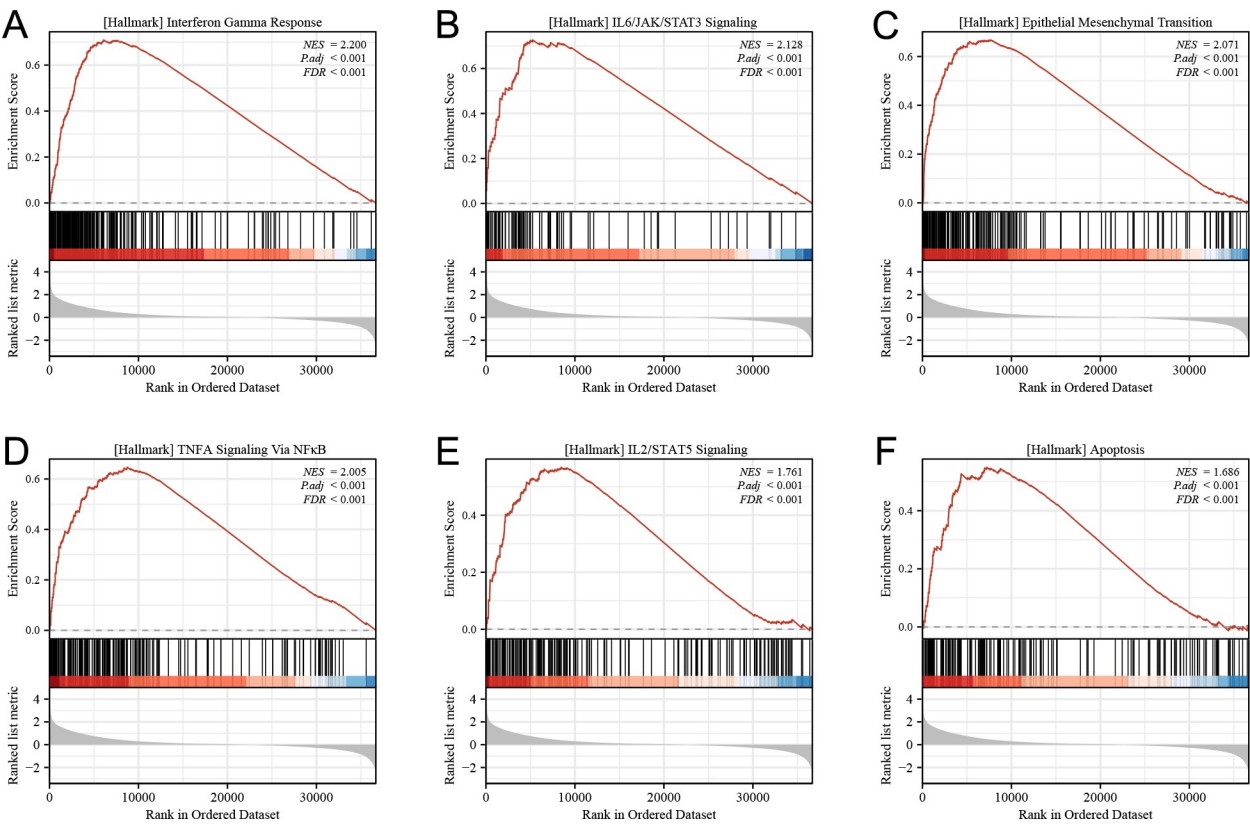

**Fig 6. Enrichment plots from gene set enrichment analysis (GSEA).**

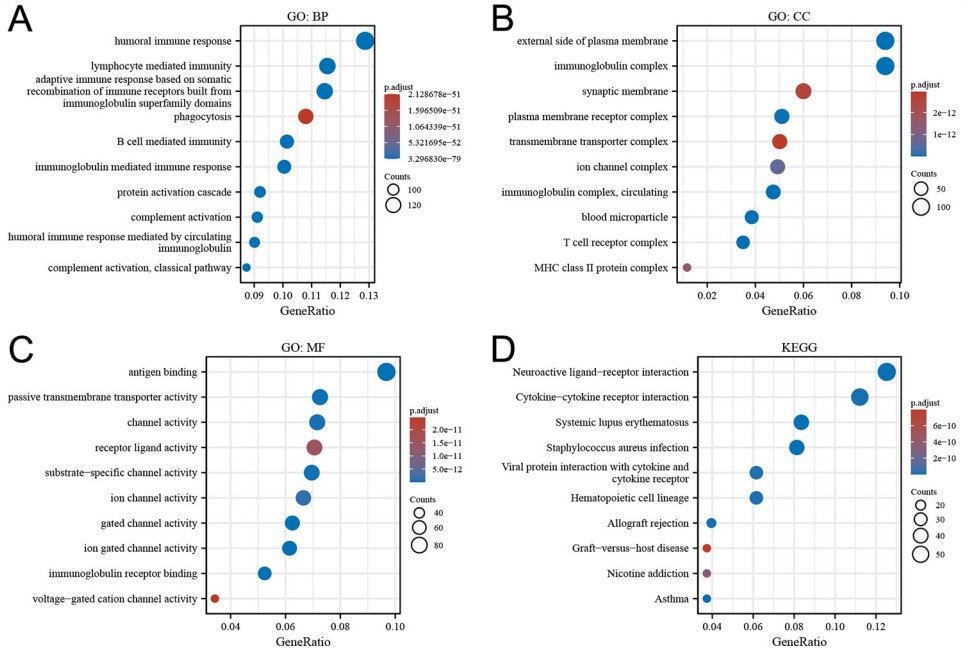

**Fig 7.** Enrichment plots from GO (A, B and C) and KEGG (D) analysis. GO, Gene Ontology; KEGG, Kyoto Encyclopedia of Genes and Genomes; BP, biological process; CC, cellular component; MF, molecular function.

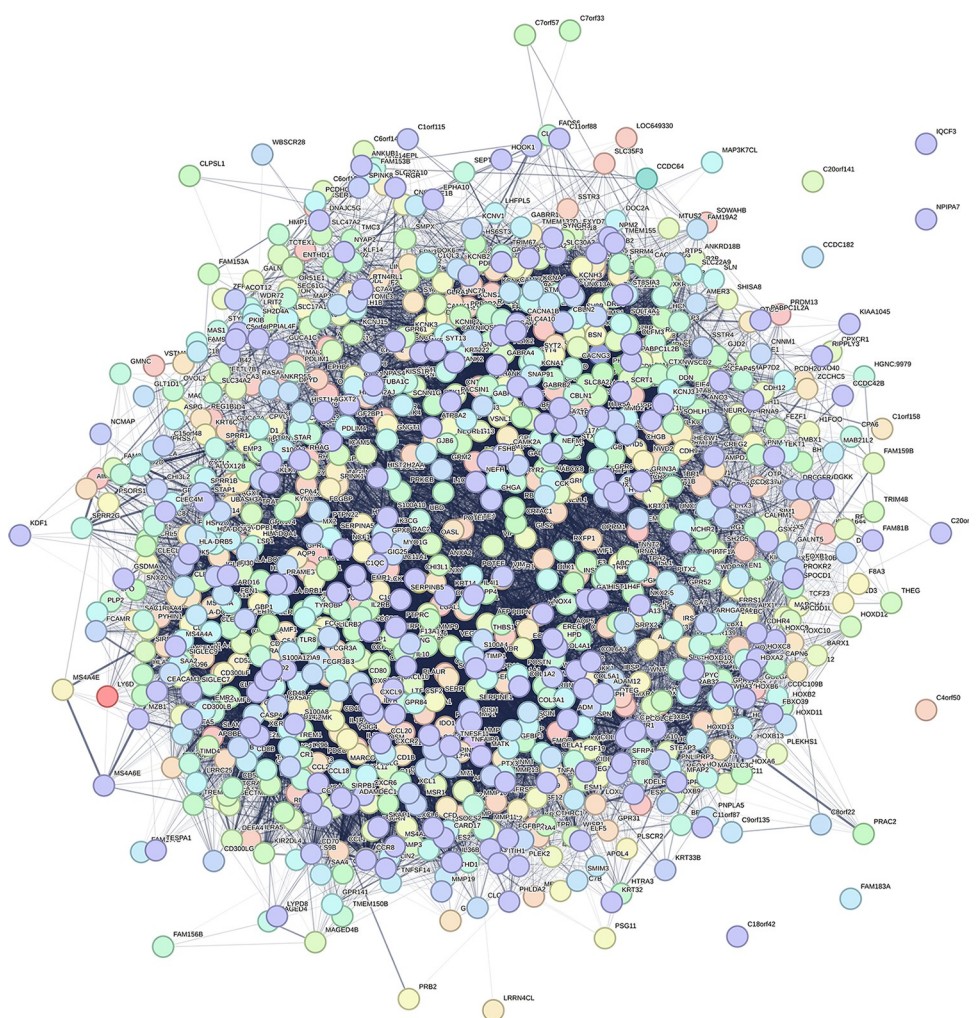

**Fig 8. Visual map of the protein-protein interaction network for high- and low-expression LINC00324 groups.**

reduce the proliferation and invasiveness of glioma cells [36]. Bioinformatics predicted that miR-375 has binding sites with LINC00324, which may regulate the proliferation, migration, and invasion of glioma cells by targeting miR-375. Thus, valid biological experiments are still needed to support our conjecture. The above results indicated that LINC00324 was associated with clinicopathological factors(WHO grading, IDH1 status, etc.) with a poorer prognosis. In addition, we found that higher expression levels of LINC00324 correlated to shorter OS and DSS time, suggesting that LINC00324 could be used as a potential prognostic biomarker for gliomas, and its role in gliomas could help us to explore the significance of LINC00324 as a target for the treatment of gliomas.

In cancer pathogenesis, lncRNAs have the intrinsic ability to interact with various molecular signals. In this context, we compared GSEA results from datasets with different LINC00324 expression levels and used them to identify the signaling pathways it differentially expresses in gliomas. The results showed that the JAK/STAT3 signaling pathway, STAT5 signaling pathway, nuclear factor-κB activation, and apoptosis significantly enriched glioma phenotypes with high LINC00324 expression. In addition, LINC00324 may be associated with cellular functions such as epithelial-mesenchymal transition and adaptive immune response. Epithelial-mesenchymal transition increases cell motility and invasiveness and leads to the distant

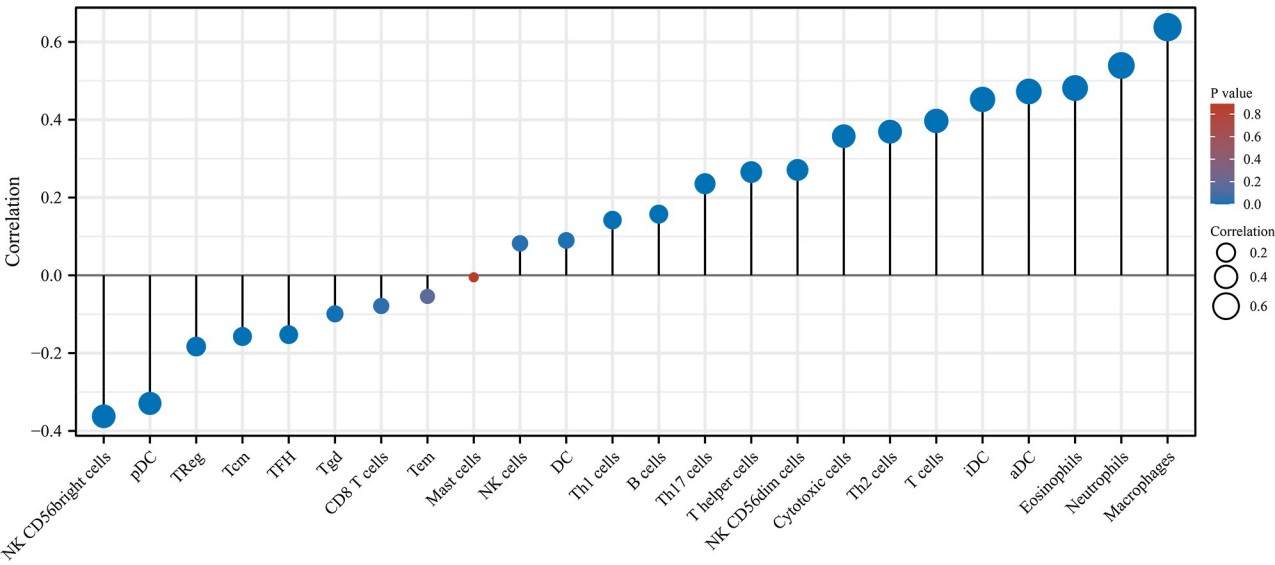

**Fig 9. Correlation between LINC00324 expression and immune infiltration.**

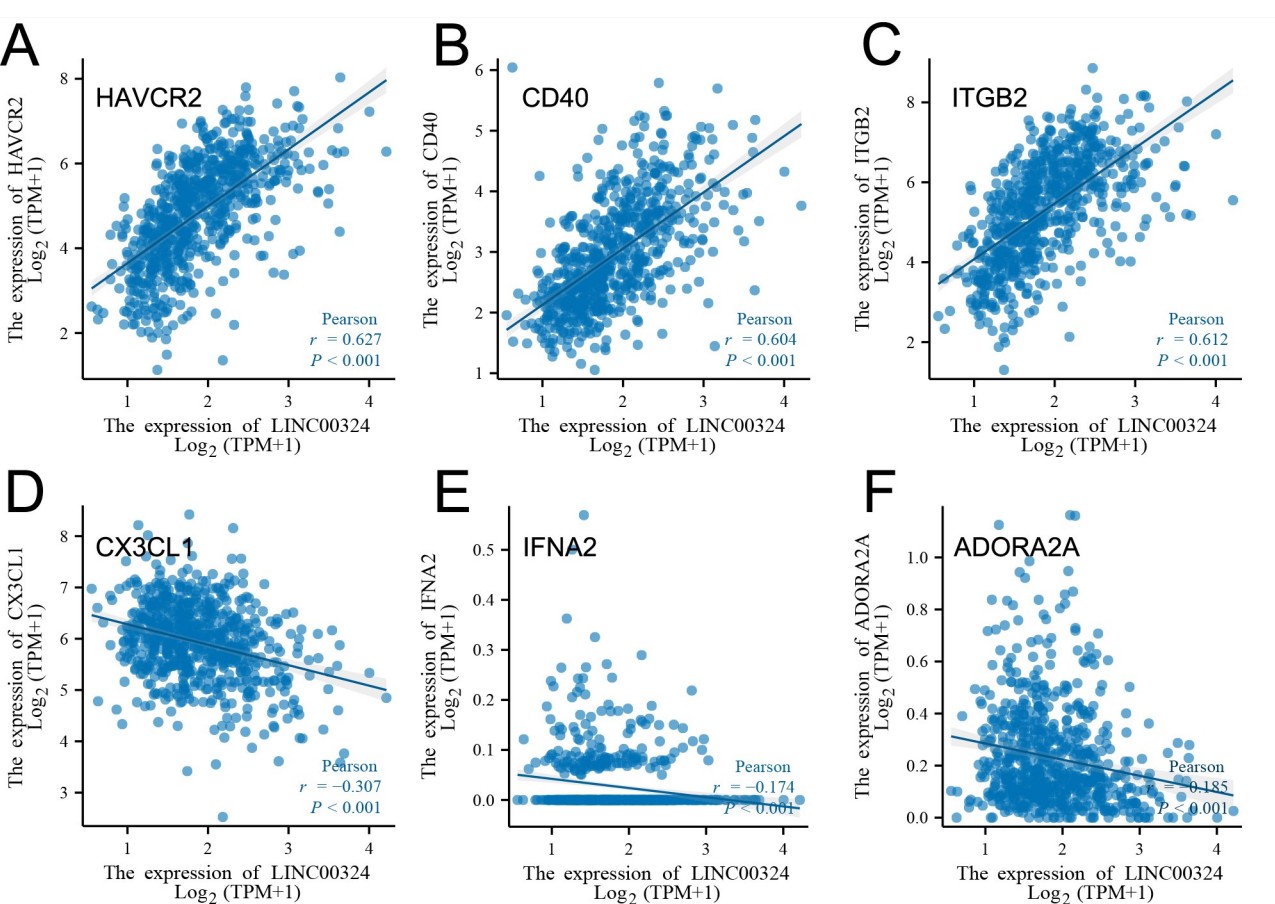

**Fig 10.** Association between LINC00324 and HAVCR2(A), CD40(B), ITGB2(C), CX3CL1(D), IFNA2(E) and ADORA2A(F) expression in glioma patients, respectively.

spread of primary tumors [37]. Recent studies have shown that mesenchymal stromal cells trigger an inflammatory response via JAK/STAT3, which promotes tumor development by secreting colony-stimulating factors, particularly granulocyte-macrophage colony-stimulating factor and macrophage colony-stimulating factor, to activate endothelial cell proliferation and migration to regulate angiogenesis [38]. The NF-κB signaling pathway, a classical signaling pathway of the inflammatory response, has been reported in several papers [39].

Subsequent studies have shown that lncRNAs are also NF-κB-interacting RNA with upregulated expression in cytotoxic T cells. The association of lncRNAs with immune cells is a complex and essential process. It has been reported in the literature that lncRNAs, including LINC00324, play an important role in immune cell activation in diseases such as hepatocellular carcinoma and lung adenocarcinoma, but the exact mechanism is not clear [40]. Our analysis also showed that LINC00324 was significantly and positively correlated with macrophages and neutrophils, consistent with the literature. CD40 is involved in transducing activating signals for inflammation and immune disorders [41], and CD40 is often expressed on dendritic cells, B cells, and macrophages. Furthermore, LINC00324 expression is positively correlated with these immune cells and strongly correlated with CD40 (r = 0.604). Similarly, HAVCR2, a gene encoding TIM-3, was positively correlated with the expression of LINC00324, which may regulate immune activity through the GALECTIN signaling pathway in the glioma immune microenvironment [42]. CX3CL1, a vital chemokine family member, has a negative regulatory role on glioma cells [43, 44]. In patients with rheumatoid arthritis, LINC00324 positively correlates with CD4+ T-cell expression and exacerbates inflammation by targeting miR-10a-5p through the NF-κB signaling pathway [45]. And whether it plays a role in gliomas is unclear. This complex interaction leads to inflammation, immune disorders, and tumor development. Therefore, based on the specific immune cells in glioma tissues and the different functional immune phenotypes, it helps to reveal the mechanism of LINC00324's role in immune regulation. These immune properties of LINC00324 prompted us to search for potential immunomodulatory factors associated with LINC00324 to provide new ideas for further exploration of immunotherapy for GBM.

Our assessment of LINC00324 mRNA expression in glioma revealed that LINC00324 plays an essential role in glioma pathogenesis, although further experimental studies are needed for confirmation. We will further expand the clinical sample size and prognostic information in the future to improve the limitations of this study. In addition to this, given that predictive characteristics were established and validated by using data from public databases, further biological evidence is needed in addition to the statistical results we provided.

## Conclusion

Our findings suggest that LINC00324 expression could serve as a potentially valuable molecular marker for predicting poor survival outcomes in glioma patients. Additionally, the JAK/STAT3 signaling pathway, NF-κB signaling pathway, and immune microenvironment disruptions appear to be key pathways involved in the regulation of LINC00324 in glioma. Our study provides new directions for glioma-specific targeted therapy. Future research in this area will provide a deeper understanding of LINC00324's role in glioma progression and its potential as a therapeutic target.

## Supporting information

**S1 Table. The correlation between LINC00324 expression and immune checkpoint-related genes.**
(DOCX)

## Author Contributions

**Conceptualization:** Shengjun Shi.

**Data curation:** Jiandong Zhu, Haoyun Yu, Kecheng Shen.

**Formal analysis:** Xin Jin, Jiandong Zhu, Haoyun Yu.

**Funding acquisition:** Jiang Wu.

**Investigation:** Jingyu Gu.

**Methodology:** Xin Jin, Jiandong Zhu.

**Resources:** Jiandong Zhu, Ziqian Yin.

**Software:** Jiandong Zhu, Ziqian Yin.

**Visualization:** Jingyu Gu, Ziqian Yin, Jiang Wu.

**Writing – original draft:** Xin Jin, Jiandong Zhu.

**Writing – review & editing:** Haoyun Yu, Zhengquan Yu.

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
