## [Decision Letter · Decision Letter 0]

27 Oct 2023

PONE-D-23-25864Dysregulation of LINC00324 promotes poor prognosis in patients with gliomaPLOS ONE

Dear Dr. Yu,

Thank you for submitting your manuscript to PLOS ONE. After careful consideration, we feel that it has merit but does not fully meet PLOS ONE’s publication criteria as it currently stands. Therefore, we invite you to submit a revised version of the manuscript that addresses the points raised during the review process.

We look forward to receiving your revised manuscript.

Kind regards,

Syed M. Faisal, Ph.D.

Academic Editor

PLOS ONE

Journal Requirements:

https://www.mdpi.com/2073-4425/13/5/851/htm

https://www.frontiersin.org/journals/oncology/articles/10.3389/fonc.2022.1039366/full

In your revision ensure you cite all your sources (including your own works), and quote or rephrase any duplicated text outside the methods section. Further consideration is dependent on these concerns being addressed

**Additional Editor Comments:**

After a thorough review by three expert reviewers, I have consolidated their feedback for your consideration.

Reviewer-1 appreciates the depth of your study on LINC00324's role in glioma. They suggest a deeper exploration into the broader implications of aberrantly expressed lncRNAs in glioma. They also recommend providing more explicit examples of how LINC00324 might influence clinical decision-making, discussing potential therapeutic implications of enriched pathways, and offering insights into LINC00324's correlation with specific immune cell types. 

Reviewer-2 recognizes the informative nature of your work but emphasizes the need for experimental evidence to support the findings. They have also highlighted areas in the manuscript that require clarity and improved presentation, such as the clear mention of LINC00324 as a long intergenic non-coding RNA and avoiding repetition in sections. 

Reviewer-3 suggests minor revisions, emphasizing the need for more detailed reporting on the selected glioma patients, increasing the number of control samples, and exploring any previous literature on LINC00324's role in other diseases. 

Given the feedback, I recommend a major revision. However, if you disagree with any of the reviewers' comments, please provide a reasoned argument in your response. We value constructive dialogue and are open to considering different perspectives to ensure the manuscript's quality and relevance.

Reviewers' comments:

Reviewer's Responses to Questions

**Comments to the Author**

1. Is the manuscript technically sound, and do the data support the conclusions?

Reviewer #1: Yes

Reviewer #2: Partly

Reviewer #3: Yes

2. Has the statistical analysis been performed appropriately and rigorously? 

Reviewer #1: Yes

Reviewer #2: Yes

Reviewer #3: Yes

3. Have the authors made all data underlying the findings in their manuscript fully available?

Reviewer #1: Yes

Reviewer #2: Yes

Reviewer #3: Yes

4. Is the manuscript presented in an intelligible fashion and written in standard English?

Reviewer #1: Yes

Reviewer #2: No

Reviewer #3: Yes

5. Review Comments to the Author

Reviewer #1: I have conducted a thorough review of the manuscript titled "Comprehensive Analysis of LINC00324 in Glioma: Expression, Clinicopathological Associations, and Implications for Prognosis and Immune Response." The study delves into the role of LINC00324 in glioma, exploring its expression patterns, associations with clinicopathological features, prognostic value, and potential impact on the immune microenvironment. While the study provides valuable insights, I have a few questions and comments that would enhance the manuscript's clarity, depth, and scientific rigor:

Introduction to LncRNAs in Gliomas:

Could the authors expand on the broader implications of aberrantly expressed lncRNAs in glioma? Highlighting specific examples of lncRNAs and their roles in glioma development and progression would contextualize the importance of LINC00324's investigation.

Study Contributions and Significance:

It would be beneficial to elaborate on the specific biological insights gained from the correlations observed between LINC00324 expression and clinicopathological factors, such as WHO grade and IDH1 status. How do these correlations enhance our understanding of LINC00324's potential role in glioma pathogenesis?

Prognostic Value and Clinical Implications:

The manuscript briefly mentions that LINC00324 could serve as a prognostic biomarker. Could the authors provide more explicit examples of how this information might influence clinical decision-making or patient management strategies in the context of glioma treatment?

Pathways and Functional Networks:

Expanding on the potential therapeutic implications of the enriched JAK/STAT3 and NF-κB signaling pathways, as well as the relationship with immune cells, would add depth to the discussion. How might targeting these pathways impact the development of novel therapeutic strategies for glioma patients?

Immune Microenvironment and Immune Response:

Could the authors provide further insights into the functional consequences of LINC00324's correlation with specific immune cell types? How might these correlations contribute to the complex immune landscape of gliomas and influence patient outcomes?

Biological Mechanisms and Further Research:

Clarification on how the interaction between LINC00324 and miR-375 may elucidate the underlying regulatory mechanisms of LINC00324 in glioma would enhance the discussion. Additionally, could the authors propose specific experimental approaches to validate the biological impact of LINC00324?

Study Limitations and Future Directions:

Given the limitations associated with utilizing public databases for predictive characteristics, it would be valuable for the authors to discuss potential strategies for integrating biological evidence with statistical results. How might future research overcome these limitations and enhance the study's findings?

Concluding Remarks and Future Implications:

Expanding on how the identification of LINC00324's role in glioma progression might influence the development of personalized therapies or diagnostics could provide a clearer roadmap for translating the study's findings into clinical applications.

I believe that addressing these questions and considerations will strengthen the manuscript's scientific rigor, depth, and relevance to the field of glioma research. Overall, the study's insights into LINC00324's potential implications for prognosis and immune response hold promise for advancing our understanding of glioma pathogenesis and treatment

Reviewer #2: The manuscript entitled "Dysregulation of LINC00324 Promotes Poor Prognosis in patients with Glioma” is an informative article. The authors have retrieved glioma patients' data from TCGA to identify the expression of long non-coding RNA LINC00324 and its involvement in the different signaling pathways.

The analyses showed that the high expression of the LINC00324 is associated with poor prognosis. The authors developed a nomogram to predict OS and DSS and have also performed GSEA revealing the differential enrichment of JAK/STAT3, EMT, STAT5, and some other pathways in the patients with the high LINC00324 expression. The high expression of the LINC00324 is also associated with the immune infiltration and expression of immune checkpoint-related genes.

Limitations: This is a completely bioinformatic analysis; no experimental evidence is provided to substantiate or prove the findings.

Although the authors have done a good job and the findings are informative and can have clinical as well as therapeutic relevance, in the absence of any experimental evidence I don’t find it fit for publication in this journal. From the pool of so much information acquired through the analyses at least, a couple of experiments could have been done to validate any of the findings. For example, the authors would have silenced or knocked out the LINCOO324 and have shown the Jak/STAT or STAT5 pathway is getting inhibited using any of the GBM cell lines.

Moreover, the article is poorly written and lacks flow, grip, and presentation.

For example:

1. The authors have started the abstract without mentioning the LINC00324 as long intergenic non-coding RNA leaving the readers to guess what it is. The authors need to mention it clearly.

2. In the materials and methods section of the abstract, the authors have written it as mRNA expression of LINC00324. Again, it gives the impression that it is not a long non-coding RNA. Authors should simply write it as “expression of LINC00324”.

3. In the background section, the authors have repeated the whole paragraph. The paragraph “LINC00324, also referred to as c17orf44, is situated on the 17p13.1 region of the human chromosome. This 2082-bp long non-coding RNA has been found to display abnormal expression patterns in various cancer types, and its dysregulation is associated with unfavorable clinical outcomes and characteristics. Notably, LINC00324 exerts influence on multiple oncogenic molecular pathways that impact crucial cellular processes such as proliferation, migration, invasion, and apoptosis. Despite this knowledge, the expression, function, and mechanism of LINC00324 in gliomas remain largely unexplored. In the current study, we conducted an analysis of LINC00324 expression and its clinical relevance in glioma. We delved into the role of LINC00324 in glioma cell proliferation and tumorigenesis and made an effort to unravel the molecular mechanisms underlying LINC00324-dependent phenotypes. Additionally, we developed a robust prognostic model aimed at determining the prognosis of glioma patients.”, this is saying almost the same thing as the previous paragraph.

4. Similarly, the discussion section has many sentences that have similar or overlapping meanings.

Reviewer #3: The manuscript entitled “Dysregulation of LINC00324 promotes poor prognosis in patients with glioma” focusses on understanding the role of LINC00324 in glioma cell proliferation and tumorigenesis, unravelling the molecular mechanisms underlying LINC00324-dependent phenotypes, developing a robust prognostic model aimed at determining the prognosis of glioma patients. The manuscript needs minor revision before it can be accepted for publication. The comments are as follows:

1. The authors have selected 699 glioma patients for the study. The authors have not reported the age group, gender of the selected patients. Also, it is important to know if the patients were smokers or alcoholic. Whether the patients had any other complication/disease beside glioma. What was the reason of glioma that is whether they have genetic history, stress, bad food habits that caused glioma. It would be great if the authors have subgroups based of age, gender, smoker/non-smoker, alcoholic/non-alcoholic, genetic/environmental of the glioma patients.

2. The authors have selected only 5 control patient samples. They should increase the control patient samples as well to at least 20.

3. Is there any previous literature that have mentioned the role of LINC00324 in glioma, any other cancer type, or and other disease?

4. The authors should also look up the exact mechanism of LINC00324 upregulation in glioma patients. However, this work is not needed for the current manuscript.

6. PLOS authors have the option to publish the peer review history of their article (what does this mean?). If published, this will include your full peer review and any attached files.

Reviewer #1: No

Reviewer #2: No

Reviewer #3: **Yes: **Sidra Islam

---

## [Author Response · Author response to Decision Letter 0]

14 Dec 2023

Response to Reviewer 1 Comments

Thank you for your decision and constructive comments on my article "Dysregulation of LINC00324 promotes in patients with glioma." (article number :PONE-D-23-25864-2560165). We have carefully considered the suggestion of Reviewer. Revision notes, point-to-point, are given as follows:

Point 1: 

Introduction to LncRNAs in Gliomas:

Could the authors expand on the broader implications of aberrantly expressed lncRNAs in glioma? Highlighting specific examples of lncRNAs and their roles in glioma development and progression would contextualize the importance of LINC00324's investigation.

Response 1:

Thanks for your suggestion. Gliomas are the most common and aggressive primary tumors of the central nervous system. In the 2016 World Health Organization (WHO) classification, innovative molecular features were included in the diagnostic criteria for gliomas based on the classification, including IDH mutations and 1p/19q co-deletions. In recent years, the emerging fields of genomics, transcriptomics, and proteomics have brought about an explosion of information about gliomas. The study of glioma biomarkers is rapidly evolving, and lncRNAs are receiving increasing attention. Recent evidence suggests that aberrant lncRNA expression plays a vital role in glioma pathogenesis, such as biogenesis, proliferation, angiogenesis, and treatment resistance. Compared with mRNAs. lncRNAs are expressed at higher levels in brain tissue than in other tissues, and thus, lncRNAs may be more suitable biomarkers for gliomas. In addition, we have not found any reports about LINC00324 in glioma research at the time of completing this manuscript, and our study has profound implications for the diagnosis and treatment of glioma. In the meantime, we have added this section to the introductory part of the manuscript.

Point 2: 

Study Contributions and Significance:

It would be beneficial to elaborate on the specific biological insights gained from the correlations observed between LINC00324 expression and clinicopathological factors, such as WHO grade and IDH1 status. How do these correlations enhance our understanding of LINC00324's potential role in glioma pathogenesis?

Response 2: 

Indeed, the primary purpose of our study on the biological mechanism of LINC00324 was to provide an experimental basis for clinical practice. From our analysis, LINC00324 was highly correlated with clinicopathologic factors; for example, the expression of LINC00324 increased with WHO classification and was highest in GBM, which has the poorest prognosis, and was higher in patients with IDH1-wildtype and 1p/19q-non codeleted (as we demonstrated in FIGURE 4). This suggests that LINC00324 is highly associated with poor prognosis. Our analysis of clinicopathological factors, on the one hand, promises future in-depth studies of LINC00324 to explore its potential as a potential prognostic biomarker for gliomas. On the other hand, these correlations provide new ideas for us to explore the specific pathogenesis of LINC00324 in gliomas, as we added in the Discussion section: LINC00324 may be associated with IDH1 mutations, involved in metabolic reprogramming and closely related to various tumor-related cellular processes.

Point 3: 

Prognostic Value and Clinical Implications:

The manuscript briefly mentions that LINC00324 could serve as a prognostic biomarker. Could the authors provide more explicit examples of how this information might influence clinical decision-making or patient management strategies in the context of glioma treatment?

Response 3: 

As mentioned above, current glioma research is increasingly focused on molecular typing and molecular pathology studies. Our results showed that LINC00324 was significantly correlated with common malignant glioma molecular phenotypes (e.g., IDH1-wildtype, 1p19q-non codeleted), and we found that high expression of LINC00324 had a poor OS and DSS time. Furthermore, our predictive results of 1-, 3-, and 5-year survival rates confirmed our suspicion that LINC00324 is prognostically important for glioma patients. Although there is still a lack of validated biological experiments to ensure this, we have reason to believe that LINC00324 and its molecular signaling pathway are expected to become new clinical therapeutic targets and prognostic markers. This also provides a new direction for our future topics and data support for exploring new glioma diagnosis and treatment strategies.

Point 4: 

Pathways and Functional Networks:

Expanding on the potential therapeutic implications of the enriched JAK/STAT3 and NF-κB signaling pathways, as well as the relationship with immune cells, would add depth to the discussion. How might targeting these pathways impact the development of novel therapeutic strategies for glioma patients?

Response 4: 

We compared GSEA results of datasets with different LINC00324 expression levels and showed that JAK/STAT3 signaling pathway nuclear factor-κB activation significantly enriched the glioma phenotype with high LINC00324 expression. Recent studies have shown that tumor cells trigger inflammatory responses via JAK/STAT3 and secrete immunomodulatory factors such as granulocyte-macrophage colony-stimulating factor and macrophage colony-stimulating factor, which activate the proliferation and migration of endothelial cells, promote tumor development, and regulate angiogenesis. The NF-κB signaling pathway is a classical signaling pathway of inflammatory response, which has been reported in several papers. Subsequent studies have shown that lncRNAs are also NF-κB-interacting RNA with upregulated expression in cytotoxic T cells. Cancer immunotherapy mainly targets immune checkpoint molecules, such as PD-1 or PD-L1, to deregulate the inhibition of conventional cytotoxic T cells and restore their anti-tumor activity. Increasing evidence suggests that lncRNAs regulate critical mechanisms of cancer immunity, from antigen presentation to T-cell depletion. Our study enriches the possibility of LINC00324 and its pathway proteins as immunotherapeutic sites, which has positive and far-reaching implications for the development of new strategies for immunotherapy in glioma patients. In the Discussion section, we also provide an extended description.

Point 5: 

Immune Microenvironment and Immune Response:

Could the authors provide further insights into the functional consequences of LINC00324's correlation with specific immune cell types? How might these correlations contribute to the complex immune landscape of gliomas and influence patient outcomes?

Response 5: 

The immune microenvironment of tumors is a complex process, and as we have previously described, there is growing evidence that lncRNAs are a crucial mechanism for regulating cancer immunity. It has been shown that lncRNAs are associated with the infiltration of NK, immature dendritic cells and mast cells, and activated B cells, as well as the expression of the immune checkpoint molecules PD-1 and PD-L1. In hepatocellular carcinoma, lincRNA-Cox2 expression mediates the reduction of IL-12, iNOS, and TNF-α in M1 macrophages, inactivating their tumor suppressor function, whereas its expression in M2 macrophages promotes cancer cell proliferation. In lung adenocarcinoma, nine types of immune cells, including macrophages, were significantly associated with lncRNAs, especially LINC00324,which is largely consistent with our findings in glioma. However, the specific role of LINC00324 in the tumor immune microenvironment remains unclear. This also provides a referable idea for us to focus our future research for deeper study.

Point 6:

Biological Mechanisms and Further Research:

Clarification on how the interaction between LINC00324 and miR-375 may elucidate the underlying regulatory mechanisms of LINC00324 in glioma would enhance the discussion. Additionally, could the authors propose specific experimental approaches to validate the biological impact of LINC00324?

Response 6: 

The interaction between LINC00324 and miR-375 may be complex. We predicted that miR-375 has a binding site with LINC00324 through biological databases and thus conjectured that LINC00324 may regulate the proliferation, migration, and invasion of glioma cells by targeting miR-375. Unfortunately, we did not find reliable biological experiments to confirm our conjecture. Therefore, we will silence or knock down LINC00324 in the future to explore the possible mechanisms of LINC00324's natural role.

Point 7: 

Study Limitations and Future Directions:

Given the limitations associated with utilizing public databases for predictive characteristics, it would be valuable for the authors to discuss potential strategies for integrating biological evidence with statistical results. How might future research overcome these limitations and enhance the study's findings?

Response 7: 

We are very grateful to you for alerting us to this critical issue. We agree that this experiment is necessary, and your suggestions provide a clear path for future research. Immediately after receiving your comments, we ordered the materials needed and prepared to improve the experiment. We mentioned this limitation in the discussion section of the article. In our following studies, we will make the necessary improvements based on your raised issues.

Point 8: 

Concluding Remarks and Future Implications:

Expanding on how the identification of LINC00324's role in glioma progression might influence the development of personalized therapies or diagnostics could provide a clearer roadmap for translating the study's findings into clinical applications.

Response 8: 

As we mentioned above, our results suggest that LINC00324 is essential in the diagnosis and treatment of gliomas, and it is not only expected to become a new molecular diagnostic and prognostic standard but also has excellent prospects in the field of immunotherapy. As we advance, we will deepen our research, extend our experiments, and make the results more applicable to the clinical setting.

We would like to express our sincere thanks to the reviewers for their enthusiastic work and hope that the revision can be recognized. Thank you again for your comments and suggestions.

Response to Reviewer 2 Comments

Point 1: 

Although the authors have done a good job and the findings are informative and can have clinical as well as therapeutic relevance, in the absence of any experimental evidence I don’t find it fit for publication in this journal. From the pool of so much information acquired through the analyses at least, a couple of experiments could have been done to validate any of the findings. For example, the authors would have silenced or knocked out the LINCOO324 and have shown the Jak/STAT or STAT5 pathway is getting inhibited using any of the GBM cell lines.

Response 1: 

We thank you for your decision and helpful comments on our manuscript. We considered the reviewers' suggestions, and this paper is a preliminary exploration of the role of LINC00324 in gliomas. We thank the reviewers for guiding our future studies, and we will further delve into the possible molecular mechanisms of LINC00324 in gliomas in our following studies.

Point 2: 

Moreover, the article is poorly written and lacks flow, grip, and presentation.

For example:

1. The authors have started the abstract without mentioning the LINC00324 as long intergenic non-coding RNA leaving the readers to guess what it is. The authors need to mention it clearly.

2. In the materials and methods section of the abstract, the authors have written it as mRNA expression of LINC00324. Again, it gives the impression that it is not a long non-coding RNA. Authors should simply write it as “expression of LINC00324”.

3. In the background section, the authors have repeated the whole paragraph. The paragraph “LINC00324, also referred to as c17orf44, is situated on the 17p13.1 region of the human chromosome. This 2082-bp long non-coding RNA has been found to display abnormal expression patterns in various cancer types, and its dysregulation is associated with unfavorable clinical outcomes and characteristics. Notably, LINC00324 exerts influence on multiple oncogenic molecular pathways that impact crucial cellular processes such as proliferation, migration, invasion, and apoptosis. Despite this knowledge, the expression, function, and mechanism of LINC00324 in gliomas remain largely unexplored. In the current study, we conducted an analysis of LINC00324 expression and its clinical relevance in glioma. We delved into the role of LINC00324 in glioma cell proliferation and tumorigenesis and made an effort to unravel the molecular mechanisms underlying LINC00324-dependent phenotypes. Additionally, we developed a robust prognostic model aimed at determining the prognosis of glioma patients.”, this is saying almost the same thing as the previous paragraph.

4. Similarly, the discussion section has many sentences that have similar or overlapping meanings.

Response 2: 

Thank you for your suggestions. However, we did invite a friend from the United States who is a native English speaker to help refine our article. We hope that the revised manuscript will be acceptable to you. Changes in the revised document are marked in red.

Response to Reviewer 3 Comments

Point 1: 

The authors have selected 699 glioma patients for the study. The authors have not reported the age group, gender of the selected patients. Also, it is important to know if the patients were smokers or alcoholic. Whether the patients had any other complication/disease beside glioma. What was the reason of glioma that is whether they have genetic history, stress, bad food habits that caused glioma. It would be great if the authors have subgroups based of age, gender, smoker/non-smoker, alcoholic/non-alcoholic, genetic/environmental of the glioma patients.

Response 1: 

We are very grateful for the reviewer's comments, which are very constructive. We provided a summary of the clinical characteristics of our selected patients (TABLE 1). Although we understand that past medical history and lifestyle habits are essential for tumor development, unfortunately, after researching the TCGA-GBM LGG database, we found that our selected patients lacked this data part, preventing us from further analysis. In addition, your suggestion reminds us of the importance of data integrity, so the collection of clinical specimens in our center has been improved after receiving your request, and we will collect enough samples and data for more in-depth research shortly.

Point 2: 

The authors have selected only 5 control patient samples. They should increase the control patient samples as well to at least 20.

Response 2:

The database we chose contained only five control samples; to increase the reliability of the data, we included 1152 normal samples from the GTEx database in the analysis and updated FIGURE 1. A note was also added to the Materials and Methods section.

Point 3: 

Is there any previous literature that have mentioned the role of LINC00324 in glioma, any other cancer type, or and other disease?

Response 3:

As we described in the preface section, " It is aberrantly expressed in various types of cancer and associated with poor prognosis and clinical features. " Some studies have shown that aberrant expression of LINC00324 is significantly related to the risk of carcinogenesis in 11 cancers. These cancers involve the human respiratory, digestive, nervous, endocrine, and reproductive systems. LINC00324 expression is upregulated in cancer tissues and cell lines of the respiratory system, including non-small cell lung cancer, nasopharyngeal carcinoma, and lung adenocarcinoma. In the digestive system, LINC00324 is highly expressed in tumor tissues and cell lines of gastric cancer, osteosarcoma, and stem cell carcinoma. In addition, high LINC00324 expression was found in colorectal cancer cell lines. In neurological tumors, LINC00324 was abnormally overexpressed in retinoblastoma. LINC00324 expression was upregulated in the endocrine system in tumor tissues and papillary thyroid cancer cell lines. However, we have not found any reports on LINC00324 in glioma studies at the time of completion of this manuscript, and our analysis has profound implications for the diagnosis and treatment of gliomas.

Point 4: 

The authors should also look up the exact mechanism of LINC00324 upregulation in glioma patients. However, this work is not needed for the current manuscript.

Response 4:

We thank you for your decision and helpful comments on our manuscript. We considered the reviewers' suggestions, and this paper is a preliminary exploration of the role of LINC00324 in gliomas. We thank the reviewers for guiding our future studies, and we will investigate possible molecular mechanisms of LINC00324 in gliomas, such as gene, protein, or signaling pathway connections, in our following studies.

---

## [Decision Letter · Decision Letter 1]

18 Jan 2024

Dysregulation of LINC00324 promotes poor prognosis in patients with glioma

PONE-D-23-25864R1

Dear Dr. Yu,

We’re pleased to inform you that your manuscript has been judged scientifically suitable for publication and will be formally accepted for publication once it meets all outstanding technical requirements.

Kind regards,

Syed M. Faisal, Ph.D.

Academic Editor

PLOS ONE

Additional Editor Comments (optional):

Reviewers' comments:

Reviewer's Responses to Questions

**Comments to the Author**

1. If the authors have adequately addressed your comments raised in a previous round of review and you feel that this manuscript is now acceptable for publication, you may indicate that here to bypass the “Comments to the Author” section, enter your conflict of interest statement in the “Confidential to Editor” section, and submit your "Accept" recommendation.

Reviewer #1: All comments have been addressed

Reviewer #2: All comments have been addressed

Reviewer #3: All comments have been addressed

2. Is the manuscript technically sound, and do the data support the conclusions?

Reviewer #1: Yes

Reviewer #2: Yes

Reviewer #3: Yes

3. Has the statistical analysis been performed appropriately and rigorously? 

Reviewer #1: Yes

Reviewer #2: Yes

Reviewer #3: Yes

4. Have the authors made all data underlying the findings in their manuscript fully available?

Reviewer #1: Yes

Reviewer #2: Yes

Reviewer #3: Yes

5. Is the manuscript presented in an intelligible fashion and written in standard English?

Reviewer #1: Yes

Reviewer #2: Yes

Reviewer #3: Yes

6. Review Comments to the Author

Reviewer #1: The manuscript has been modified according to the suggestions and should be accepted as revised version.

Reviewer #2: The authors have made good effort in addressing the comments. The current form the manuscript is in good shape for acceptance.

Reviewer #3: The manuscript entitled “Dysregulation of LINC00324 promotes poor prognosis in patients with glioma” analyzes the expression and clinical relevance of LINC00324 in glioma. The authors have investigated the role of LINC00324 in glioma cell proliferation and tumorigenesis. The authors have attempted to elucidate the molecular mechanisms associated with LINC00324-dependent phenotypes. The authors have constructed a reliable prognostic model to determine the prognosis of glioma patients. All the comments have been addressed and the manuscript can proceed forward for submission.

7. PLOS authors have the option to publish the peer review history of their article (what does this mean?). If published, this will include your full peer review and any attached files.

Reviewer #1: No

Reviewer #2: No

Reviewer #3: **Yes: **Sidra Islam

---

## [Editor Report · Acceptance letter]

16 Mar 2024

PONE-D-23-25864R1 

PLOS ONE

Dear Dr. Yu, 

I'm pleased to inform you that your manuscript has been deemed suitable for publication in PLOS ONE. Congratulations! Your manuscript is now being handed over to our production team.

Kind regards, 

on behalf of

Dr. Syed M. Faisal 

Academic Editor

PLOS ONE